# Improving SARS-CoV-2 variants monitoring in the absence of genomic surveillance capabilities: a serological study in Bolivian blood donors in October 2021 and June 2022

Lucia Inchauste[1]*, Elif Nurtop[1], Lissete Bautista Machicado[2], Yanine Leigue Roth[3], Shirley Lenz Gonzales[4], Maria Luisa Herrera[5], Katty Mina Villafan[6], Pedro Mamani Mamani[7], Marcelo Ramos Espinoza[8], Juan Carlos Pavel Suarez[9], Juan Cansio Garcia Copa[10], Yitzhak Leigue Zabala[11], Etzel Arancibia Cardozo[12], Pierre Gallian[1,13], Xavier de Lamballerie[1], Stéphane Priet[1]*

[1]Unité des Virus Émergents, Marseille, France; [2]Programa Nacional de Sangre, Ministerio de Salud y Deportes, La Paz, Bolivia; [3]Banco de Sangre de Referencia Departmental de Beni, Beni, Bolivia; [4]Banco de Sangre de Referencia Departmental de Chuquisaca, Sucre, Bolivia; [5]Banco de Sangre de Referencia Departmental de Cochabamba, Cochabamba, Bolivia; [6]Banco de Sangre de Referencia Departmental de El Alto, El Alto, Bolivia; [7]Banco de Sangre de Referencia Departmental de La Paz, La Paz, Bolivia; [8]Banco de Sangre de Referencia Departmental de Oruro, Oruro, Bolivia; [9]Banco de Sangre de Referencia Departmental de Pando, Pando, Bolivia; [10]Banco de Sangre de Referencia Departmental de Potosí, Potosí, Bolivia; [11]Banco de Sangre de Referencia Departmental de Santa Cruz, Santa Cruz, Bolivia; [12]Banco de Sangre de Referencia Departmental de Tarija, Tarija, Bolivia; [13]Etablissement Français du Sang, La Plaine Saint Denis, France

**\*For correspondence:**
lucia-paola.inchauste-jordan@
univ-amu.fr (LI);
stephane.priet@univ-amu.fr (SP)

**Competing interest:** The authors declare that no competing interests exist.

## eLife Assessment

This serostudy of blood donors in Bolivia (a country with very high COVID death rates in 2020-21) provides **useful** insights on the successive viral variants of SARS-CoV-2 over 2021 and 2022. Using **compelling** antibody and neutralization assays, the authors describe variant specific distributions in the different parts of Bolivia. The main methodological advance is to use serology to understand variant diversity, which in turn helps deepen understanding of "hybrid" immunity from widespread infection (and vaccination).

## Abstract

**Background:** Unlike genomic data, serological data have not been previously leveraged to evaluate the SARS-CoV-2 variants circulation. In Bolivia, sustained genomic surveillance capacities were lacking, especially at the beginning of the pandemic.
**Methods:** In 2021 and 2022 we estimated the prevalence of anti-SARS-CoV-2 antibodies in Bolivian blood donors and explored the feasibility of using virus serum neutralization data for variants thought to have circulated to map their circulation across all departments over a year-long follow-up period. Anti-S1 and anti-NCP SARS-CoV-2 IgGs were studied, along with

virus neutralization tests for ancestral-D614G, Gamma, Delta, and Omicron BA.1 lineages of SARS-CoV-2.

**Results:** Between 2021 and 2022, the overall prevalence of anti-S1 and anti-NCP antibodies increased, reaching values over 90%, demonstrating that a large proportion of the Bolivian population was no longer naïve to the virus. Viral neutralization data, analyzed through multiple approaches, revealed the spread of the Gamma variant up to 2021, particularly impacting northern departments. In 2022, Gamma continued to circulate in southernmost departments of the country, and the emergence of Omicron BA.1 was detected. These trends align with publicly available genomic data from neighboring countries.

**Conclusions:** Our serological analyses successfully identified both new antigenic groups, such as Omicron BA.1, and individual variants related to previously circulating groups, such as Delta. The study contributes insights into overall population immunity to SARS-CoV-2 and variant-specific immunity levels across different regions of Bolivia. It also emphasizes the potency of seroprevalence studies in informing public health decisions and underscores their value in capturing the initial phases of emerging epidemics when variant diversity is limited, facilitating timely genomic surveillance setup.

**Funding:** This study was supported by the French National Research Institute for Sustainable Development (IRD), the project EMERGEN-PRI #22275 of the ANRS I MIE (INSERM), and the European Union's Horizon 2020 research and innovation program (European Virus Archive Global, grant agreement No. 871029). The funders of the study had no role in study design, data collection, data analysis, data interpretation, or writing of the report.

## Introduction

Low- and middle-income countries in the Latin American region are vulnerable to health crises due to a combination of factors such as poverty, fragile political and health systems, persistent and pervasive inequality in income, health care, and education, among others (*The Lancet, 2020*). The SARS-CoV-2 pandemic started in South America with the detection of the first case on February 26, 2020, at São Paulo, Brazil (*Melo et al., 2020*). Although it was initially thought that the pandemic would have less of an impact on this region due to its younger population structure, the pandemic eventually had a significant impact, accounting for 25% of infections worldwide (*The Lancet, 2021*).

Seroprevalence studies are a valuable adjunct to active surveillance because they allow analysis of the level of immunity of a population to a specific pathogen without the need for prospective testing and also provide information on the frequency of cases that do not attract medical attention (asymptomatic infections) (*Wu and Riley, 2014*). During the first years of the SARS-CoV-2 pandemic, a few South American population-based seroprevalence studies were conducted in Colombia, Perú, Chile, and Brazil (*Reyes-Vega et al., 2021*; *Álvarez-Antonio et al., 2021*; *Vial et al., 2022*; *Mercado-Reyes et al., 2022*; *Nicolete et al., 2022*; *Buss et al., 2021*; *Prete et al., 2022*). Reported seroprevalence values varied through time and geographical location. Notably, high prevalence was observed in locations such as Iquitos-Perú (70%) or Manaus-Brazil (54%) after the first wave (*Álvarez-Antonio et al., 2021*; *Buss et al., 2021*). However, an unexpected violent second wave occurred concomitantly with the emergence of the Gamma variant, despite the high levels of seroprevalence reported earlier (*Buss et al., 2021*; *Prete et al., 2022*; *Sabino et al., 2021*). Despite the serious impact of COVID-19 on public health in Bolivia, research studies are still lacking: one study reported a seroprevalence of 43.4% in January 2021 among healthcare workers in the city of Cochabamba (*Saba Villarroel et al., 2022*), but no nationwide seroprevalence study is currently available.

To date, the circulation of SARS-CoV-2 variants has mainly been studied through molecular surveillance, giving the proportion of circulating variants in real time. Therefore, genomic surveillance and serology offer distinct yet complementary insights thus far. However, in countries without significant genomic surveillance capabilities at the start of the epidemic, it has become difficult to assess the spread and impact of variants. Here, we have attempted to reconstruct parts of the history of the spread of SARS-CoV-2 in Bolivia by studying the prevalence and specific neutralizing capacity of anti-SARS-CoV-2 antibodies in blood donors from all Bolivian departments after the second and third COVID-19 waves. Based on the results obtained, we discuss the potential contribution of seroprevalence and

seroneutralization studies of viral variants and their complementarity with viral genomics data during the different phases of dissemination of a pathogen such as SARS-CoV-2.

## Methods

### Background

In Bolivia, the first case of SARS-CoV-2 was reported on March 10, 2020, and to date, 1,209,619 cases and 22,407 deaths have been reported (**WHO, 2023**). As for the rest of South America, Bolivia has experienced five waves of SARS-CoV-2 cases: the initial wave (May-September 2020), followed by a second (January-July 2021), a third (December 2021 to February 2022), a fourth (June-August 2022) and the most recent (November 2022 to January 2023). Data on variants circulating in each wave in South America is available in several free-access databases (e.g., the PAHO dashboard) but Bolivian genomic data are scarce and only a few sequences have been deposited (**PAHO, 2023**; **GISAID, 2023**; **Hodcroft, 2023**).

During the first wave, COVID-19 cases in Bolivia neighboring countries were due to ancestral and D614G SARS-CoV-2 strains (**Figure 1—figure supplement 1**; **PAHO, 2023**; **GISAID, 2023**; **Hodcroft, 2023**). During the second wave, Alpha, Gamma, Lambda, and Mu variants were the main types identified, with variable distribution among countries. In contrast to Europe, the Alpha variant was not the most prevalent in any country of South America. Moreover, no specific wave was produced, and no associated increase in mortality was observed due to the circulation of the Delta variant. The start of the third wave, as in the rest of the world, was concomitant with the emergence and spread of the Omicron (BA.1) variant. The next two waves were mainly caused by the Omicron variants BA.2/BA.5 and BQ.1/XBB, respectively.

### Specimens and ethical considerations

Samples were collected from volunteer blood donors who agreed to participate in the study and provided written consent. The study was approved by the ethics committee of the Dr. Mario Ortíz Suárez hospital, Santa Cruz de la Sierra, Bolivia (N° FWA0002686) and by the National Blood Program from the Ministry of Health and Sports of Bolivia. Sampling was conducted in close collaboration with the National Blood Program and each Departmental Reference Blood Bank. A first batch of 4238 serum samples was collected between August 26 and October 04, 2021. Samples were obtained from all Departmental Reference Blood Banks of Bolivia, which included the cities of Beni (N=733), Chuquisaca (N=294), Cochabamba (N=500), La Paz (N=500), El Alto (N=500), Oruro (N=500), Pando (N=414), Potosí (N=197), Santa Cruz (N=600), and Tarija (N=500). A second batch was collected between May 30 and June 23, 2022, including a total of 1161 sera from the Departmental Reference Blood Banks of Cochabamba (N=212), La Paz (N=264), Santa Cruz (N=186), and Tarija (N=499).

### Vaccination coverage

Vaccination campaign against SARS-CoV-2 in Bolivia started on January 29, 2021 (**Bolivia MdSyDd, 2021a**). Following the SAGE recommendations, vaccination started for the healthcare workers, then extended to the elderly and people living with underlying health conditions and later expanded to the general population. Vaccines distributed in Bolivia included the mRNA vaccines Comirnaty (Pfizer-BioNTech) and mRNA-1273 (Moderna), the inactivated virus vaccines CoronaVac (Sinovac Biotech, China) and BBIBP-CorV (Sinopharm, China), the (adeno)viral vector vaccines Ad26.CoV2.S (Johnson & Johnson), Vaxzevria (AstraZeneca), and Sputnik V (Gamaleya Research Institute of Epidemiology and Microbiology). Administration of a booster dose was started on October 11, 2021 (**Bolivia MdSyDd, 2021b**).

SARS-CoV-2 vaccination coverage was estimated for both sampling periods (October 27, 2021 and May 30, 2022) based on data reported by the Bolivian Ministry of Health and the Bolivian National Institute of Statistics (**Bolivia MdSyDd, 2022**; **Bolivia MdSyDd, 2021a**; **INE, 2023**). To calculate vaccination coverage, the number of vaccine doses administered by the Bolivian Ministry of Health was related (by department and nationally) to the total eligible population according to the Bolivian National Institute of Statistics. At the time of the first and second sampling periods, the eligible population included individuals over 18 years of age and over 11 years of age, respectively.

## Cell lines

Vero E6 TMPRSS2 + cells were provided by the NIBSC Research Reagent Repository, UK (Cat# 100978). Vero E6 TMPRSS2 + were not further authenticated but were confirmed as mycoplasma free by qPCR as described previously (*Störmer et al., 2009*).

## Immunoassays

All serum samples were first screened using the commercial kit Anti-SARS-CoV-2 Quantivac ELISA IgG (Euroimmun) according to manufacturer's recommendations to detect anti-Spike S1 antibodies. Specimens with BAU/mL <25.6, 25.6≤BAU/mL <35.2, and BAU/mL ≥35.2 were considered negative, equivocal, and positive, respectively.

Samples were also tested for the presence of anti-NCP SARS-CoV-2 IgG using an in-house qSAT assay, as described elsewhere (*Inchauste et al., 2024*). Briefly, the CTD domain of the SARS-CoV-2 N protein (EVAg, ref: 100 P-03957) was coupled to a MAGPLEX magnetic microsphere (Luminex Corporation) using the xMAP Antibody Coupling Kit (Luminex Corporation) following manufacturer's recommendations (60 pmol/$10^6$ beads). The coupled beads were resuspended and counted on a Countess II Automated Cell Counter (Thermo) to a final concentration of $2 \times 10^6$ beads/ml. Samples diluted in Wash Solution (Thermo) at 1/400 were incubated with 1,000 coupled beads per well for 1 h at room temperature in a plate shaker protected from light. After two washes, the beads were incubated with R-Phycoerythrin AffiniPure F(ab')$_2$ Fragment Goat Anti-Human IgG F(ab')$_2$ (Jackson ImmunoResearch, cat#: 109-116-097) for 1 hr at room temperature in a plate shaker protected from light. After washing, antigen-antibody reactions were read on an INTELLIFLEX system (Luminex Corporation,) and the results were expressed as median fluorescence intensity (MFI). Cut-off values for each SARS-CoV-2 antigen for the S, RBD, and NCP were described elsewhere (*Inchauste et al., 2024*). The sensitivity and specificity of the in vitro assays were described previously (*Inchauste et al., 2024*).

For virus Neutralization Tests (VNT), strains from the B.1 (Bav-Pat1/2020 strain [D614G]), P.1 (SARS-CoV-2/2021/JP/TY7-503 strain [Gamma]), B.1.617.2 (SARS-CoV-2/2021/FR/0610 strain [Delta]), and B.1.1.529 (UVE/SARS-CoV-2/2021/FR/1514 strain [Omicron BA.1]) lineages of SARS-CoV-2 were used. Except for the Bav-Pat1/2020 strain (courtesy of Prof. Drosten, Berlin), all strains were provided by the European Virus Archive (EVAg) repository (https://www.european-virus-archive.com/). VNTs were performed for samples with a positive or equivocal result in the anti-Spike S1 ELISA as previously described (*Gallian et al., 2020*). Four separate VNT assays were performed in duplicate with strains representative of the D614G, Gamma, Delta, and Omicron BA.1 variants. Briefly, the test consisted of mixing 110 µL of a serially diluted (from 1/20 to 1/40960) patient serum sample (in DMEM with 1% of penicillin/streptomycin, Non-Essential Amino Acids and Glutamine), with 110 µL of a fixed quantity of a given SARS-CoV-2 strain, to reach a final concentration of 0.5 TCID$_{50}$/µL of plasma dilution. This mixture was incubated for 1 hr at 37 °C. One hundred µL was then transferred onto a confluent Vero E6 TMPRSS2 + cells monolayer and incubated at 37 °C under 5% CO$_2$. On day 5 post-infection, dilutions showing a cytopathic effect (CPE) were considered negative (no neutralization), and those without CPE were considered positive (complete neutralization of the SARS-CoV2 inoculum). The neutralization titer referred to as the highest serum dilution with a positive result. Specimens with a VNT titer ≥ 20 were considered positive.

## SARS-CoV-2 seroprevalence

Prevalence values for anti-S1 and anti-NCP antibodies, and for neutralizing antibodies (nAbs) against the ancestral D614G (D614G), Gamma (γ), Delta (δ), and Omicron BA.1 (o) variants were calculated using the Prevalence package RStudio version R-4.0.3 (*Devleesschauwer et al., 2013*). Positive or negative status for each sample and every variable was determined using the cut-offs described above. For all variables, the number of positive and negative results was used for prevalence computing.

Regarding nAbs, prevalence values for each variant (D614G, γ, δ, o) in each department and nationwide were calculated using the regular seropositivity titer cut-off at 20 (samples with a titer ≥20 were considered positive) and an alternative "high level" seropositivity titer cut-off at ≥640.

## Evaluation of the circulating variants using a 'VNT Titers ratio method'

The circulation of variants in Bolivia before October 2021 and June 2022 was first analyzed through the distribution of samples exhibiting titers higher than the D614G ancestral strain for a given variant

of interest (γ, δ, ο). The base 2 logarithm of VNT titer ratios (log2-ratios) for the following variant pairs was calculated: D614G/Gamma (D614G/γ), D614G/Delta (D614G/δ), D614G/Omicron BA.1 (D614G/ο). Negative log2-ratio indicated that the variant in the denominator (γ, δ, ο) had a higher titer than D614. A log2-ratio of 0 indicated that both variants presented the same titer. To evaluate the circulation level of γ, δ, and ο variants under increasingly stringent conditions, we calculated the proportion of the population with log2-ratio values of ≤0 (variant titer equal to or greater than D614G), ≤−1 (variant titer at least twice that of D614G), and ≤−2 (variant titer at least four times that of D614G) for each department.

## Quantification of the circulating variants according to antigenic groups

Antigenic mapping of SARS-CoV-2 based on the neutralizing activity of each variant (*Mykytyn et al., 2022*; *Wilks et al., 2023*; *van der Straten et al., 2022*) enables us to group the ancestral variants D614G, Alpha, Delta, Lambda, and Epsilon in one group (GI), Beta, Gamma, Mu, Iota, and Zeta in a second group (GII), and Omicron and its derivatives into a third group (GIII). To initiate the quantification of the variants circulating in Bolivia, we initially employed a straightforward approach to estimate the percentage of samples in each antigenic group within each department of Bolivia using two different methods (the 'Average titer method' and the 'Variant assignment method').

### Average titer method

To assess the neutralizing response across distinct antigenic groups, we used the average VNT titer for the D614G and Delta variants to delineate antigenic group I (GI). Antigenic group II (GII) was established based on the VNT titer for the Gamma variant, while antigenic group III (GIII) was defined by the VNT titer for the Omicron BA.1 variant. For evaluating the percentage of each antigenic group within different regions of Bolivia, GI was considered dominant if its titer surpassed those of GII and GIII. Similarly, GII was considered if its titer exceeded those of GI and GIII, and GIII was considered if its titer equaled or surpassed those of GI and GII. Additionally, the prevalences of GI (from ancestral, Alpha, Delta, Lambda, and Epsilon variants), the prevalences of GII (from Beta, Gamma, Mu, Iota, and Zeta variants), and the prevalences of GIII (from all Omicron-derived variants) were obtained for neighboring countries of Bolivia using publicly accessible genomic data when available.

### Variant assignment method
#### Assignment rules

We implemented several rules taking into account the previously described decrease in neutralizing activity of variants *Mykytyn et al., 2022*; *Wilks et al., 2023*; *van der Straten et al., 2022* to determine which of the four variants tested was expected to be responsible for the main or last infection (*Figure 3—figure supplement 1*). Main rules relied on D614G/γ, D614G/δ, and D614G/ο log2-ratios to differentiate between D614G, Gamma, Delta, and Omicron variants. Given that Tarija had the lowest variant circulation level and the highest vaccination coverage among the departments analyzed as of October 2021, this department was chosen as a reference for establishing the D614G cut-off values for the subsequent rules defining the infection by γ, δ, and ο variants. Given that approximately 90% of the population in this department exhibited D614G/γ, D614G/δ, and D614G/ο log2-ratios ≥0, ≥0, and ≥1 respectively, the D614G variant was assigned to compliant samples. If any of these log2-ratios fell below these threshold values, the corresponding variant (γ, δ, or ο) was assigned. Secondary rules were applied on a fraction of samples when the log2-ratios for two pairs were below the threshold values. Additional log2-ratios for Delta/Gamma (δ/γ), Gamma/Omicron BA.1 (γ/ο), and Delta/Omicron BA.1 (δ/ο) pairs were thus considered to differentiate between the two variants. If both the D614G/γ and D614G/δ log2-ratios were ≤ −1, a δ/γ log2-ratio≤0 was chosen to indicate a Gamma infection, while a δ/γ log2-ratio≥1 indicated a Delta infection. When the D614G/ο log2-ratio was ≤0 and the D614G/γ log2-ratio was also ≤ −1, a γ/ο log2-ratio≥1 indicated a Gamma infection; otherwise (γ/ο log2-ratio≤0), Omicron BA.1 was assigned. Similarly, if the D614G/ο log2-ratio was ≤0 and the D614G/δ log2-ratio was ≤ −1, a δ/ο log2-ratio≥1 suggested a Delta infection; otherwise (δ/ο log2-ratio≤0), Omicron BA.1 was assigned.

Quantification according to assignment rules

Variants assigned as D614G and Delta were grouped in the antigenic group I, Gamma was set as the antigenic group II, and Omicron BA.1 as the antigenic group III. The percentage of samples in each antigenic group was calculated for all departments for each sampling period.

## Quantification of the circulating variants according to individual variants

The percentage of samples assigned through the Assignment Rules (*cf. supra*) to D614G, Gamma, Delta, or Omicron BA.1 was calculated for all departments for each sampling period.

## Results

### Population studied

A first sampling was carried out in blood donors from all Bolivian departments from August 26 to October 4, 2021, after the second epidemic wave (*Figure 1—figure supplement 1*). A second sampling to follow the evolution of the virus 6 months later was carried out in the 3 most populated departments as well as in the most southern department (May 30 to June 23, 2022, corresponding to the phase before the fourth wave).

Among the 4738 blood donors included for the first period, age ranged from 18 to 64 years, with a median of 29 years (IQR, [23-37]). In the second period, 1161 blood donors were included, ranging in age from 18 to 82 years, with a median of 32 years (IQR, [25-41]). There was a balanced distribution of males (41.9% and 44.4%) and females (58.1% and 55.0%) for both periods.

### Vaccination coverage

Vaccination coverage data obtained from the Bolivian Ministry of Health are summarized in *Figure 1—figure supplement 2* for both periods with respect to vaccination coverage for the first dose, two doses and a booster (three doses).

In October 2021, the vaccination coverage of the Bolivian population with a first dose ranged from 47.6% (Potosí) to 70.3% (Tarija), reaching 59.3% at the national level. Coverage for the complete vaccination schedule (referred to as two doses) varied from 38.4% (Pando) to 61.4% (Tarija), with a national average of 50.1%. The proportion of people who received one boost was very low, ranging from 0% (Pando) to 4.5% (Cochabamba), with a national average of 3.2%.

In May 2022, the coverage for one dose increased significantly, ranging from 62.3% (Pando) to 84.3% (Tarija), with a national average of 76.5%. Full vaccination coverage (2 doses) reached 46.6% (Pando) to 72.9% (Tarija), or 63.8% nationally. The one boost coverage reached 16.2% nationwide (10.9% [Beni] to 22.4% [Tarija]). Compared to the first sampling period, regions with higher (Tarija) and lower (Pando) vaccine coverage remained the same.

### Anti-SARS-CoV-2 Spike S1 and -NCP antibody prevalence

The anti-SARS-CoV-2 Spike S1 (anti-S1) antibody levels were assessed in each department for both sampling periods (*Figure 1A and C*).

In October 2021, at the national level, anti-S1 seroprevalence was 86.3% (95%CI: 85.3–87.6) and varied from 77.2% (Beni) to 92.4% (Potosí). Seroprevalence increased to 97.1% (95%CI: 95.9–97.9) by June 2022, ranging from 96.6% (Tarija) to 98.5% (La Paz).

In October 2021, the anti-NCP seroprevalence was 60.8% (95%CI: 59.5–62.1) at a national level, increasing to 90.2% (95%CI: 88.3–91.7) in June 2022 (*Figure 1B and D*). In 2021, anti-NCP seroprevalence spanned from 52.7% (Pando) to 76.1% (Potosí). In 2022, the range extended from 85.5% (Santa Cruz) to 92.8% (Tarija). The anti-NCP antibody prevalence was always lower than anti-S1 antibody prevalence in October 2021 and June 2022. The anti-NCP increase in the four regions tested was around 30%, while the anti-S1 increase was 8–13%.

The estimated hybrid immunity, based on the prevalence of anti-S1 and anti-NCP antibodies, ranged from 51.4% in Pando to 73.6% in Potosí in 2021. By 2022, this increased to between 83.3% in Santa Cruz and 90.6% in Tarija.

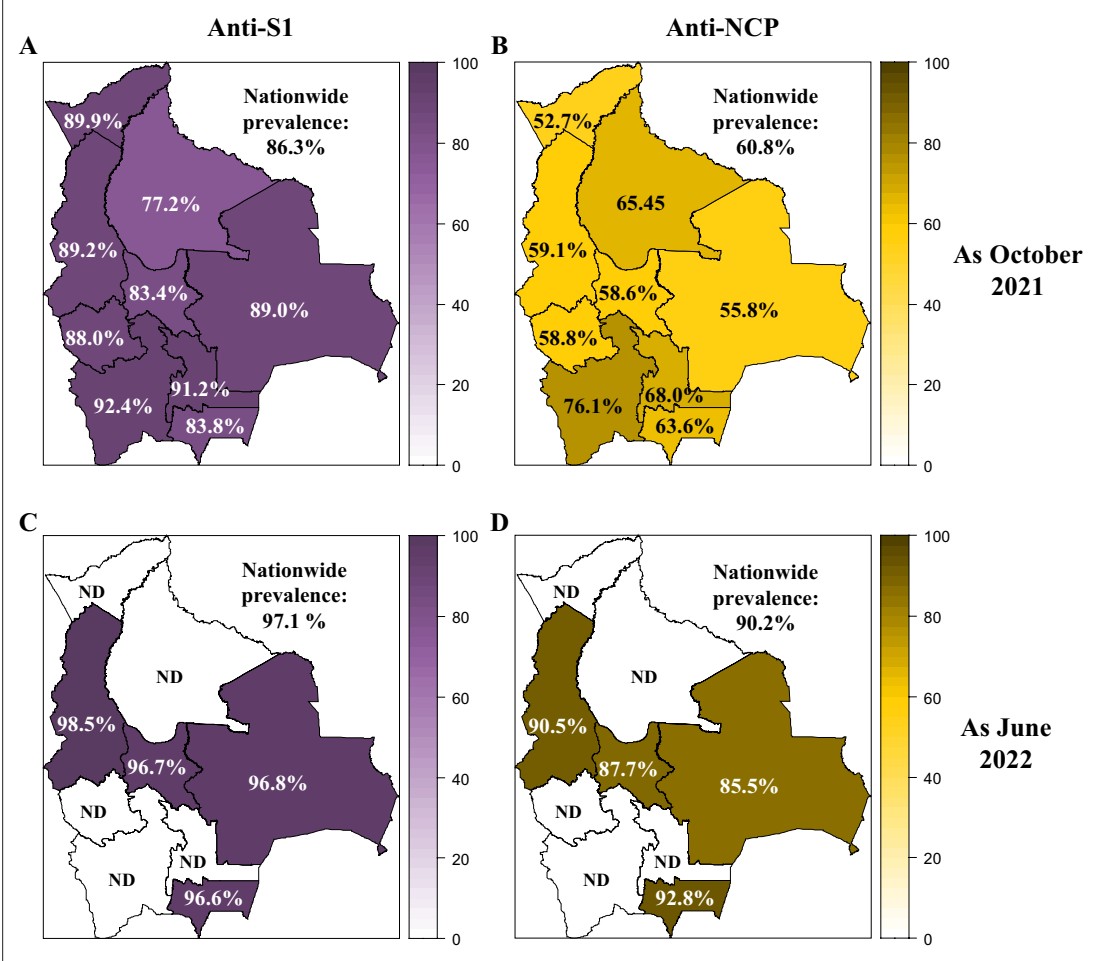

**Figure 1.** Prevalence of anti-SARS-CoV-2 Spike S1 and -NCP antibodies in Bolivia as of October 2021 and June 2022. The prevalence of anti-SARS-CoV-2 Spike S1 (**A, C**) and -NCP (**B, D**) antibodies was reported for each department in Bolivia and nationwide as of October 2021 (**A, B**) and June 2022 (**C, D**).

The online version of this article includes the following figure supplement(s) for figure 1:

**Figure supplement 1.** SARS-CoV-2 associated morbidity and mortality, and variant circulation in Bolivian surrounding countries (Argentina, Brazil, Chile, and Perú) in 2021 and 2022.

**Figure supplement 2.** Vaccination coverage in Bolivia as of October 2021 and May 2022.

## Prevalence of neutralizing antibodies

### Prevalence of nAbs against D614G

In October 2021, the nationwide prevalence of D614G nAbs at cut-offs ≥20 and ≥640 were 83.1% and 29.1% (*Figure 2*, *Figure 2—figure supplement 1*), respectively, with a geometric mean titer (GMT) of 159 (*Figure 2—figure supplement 2*). The prevalence of D614G nAbs using the cut-off ≥20 was consistently high across the entire country, ranging from 76.8% in Cochabamba to 90.3% in Potosí. Increasing the cut-off to ≥640 revealed that Pando and Tarija had the highest (45.6%) and lowest (17.7%) prevalence, respectively. The rest of the country showed values ranging between 21.5% and 34.9%.

By June 2022, D614G nAbs nationwide prevalence increased to 98.5% and 57.7% for the cut-off ≥20 and ≥640 respectively, with a GMT of 556. Prevalence obtained with the cut-off ≥20 was high across all tested departments, nearly reaching saturation. Using the cut-off ≥640, the highest prevalence was observed in La Paz (73.2%) and the lowest in Tarija (42.9%).

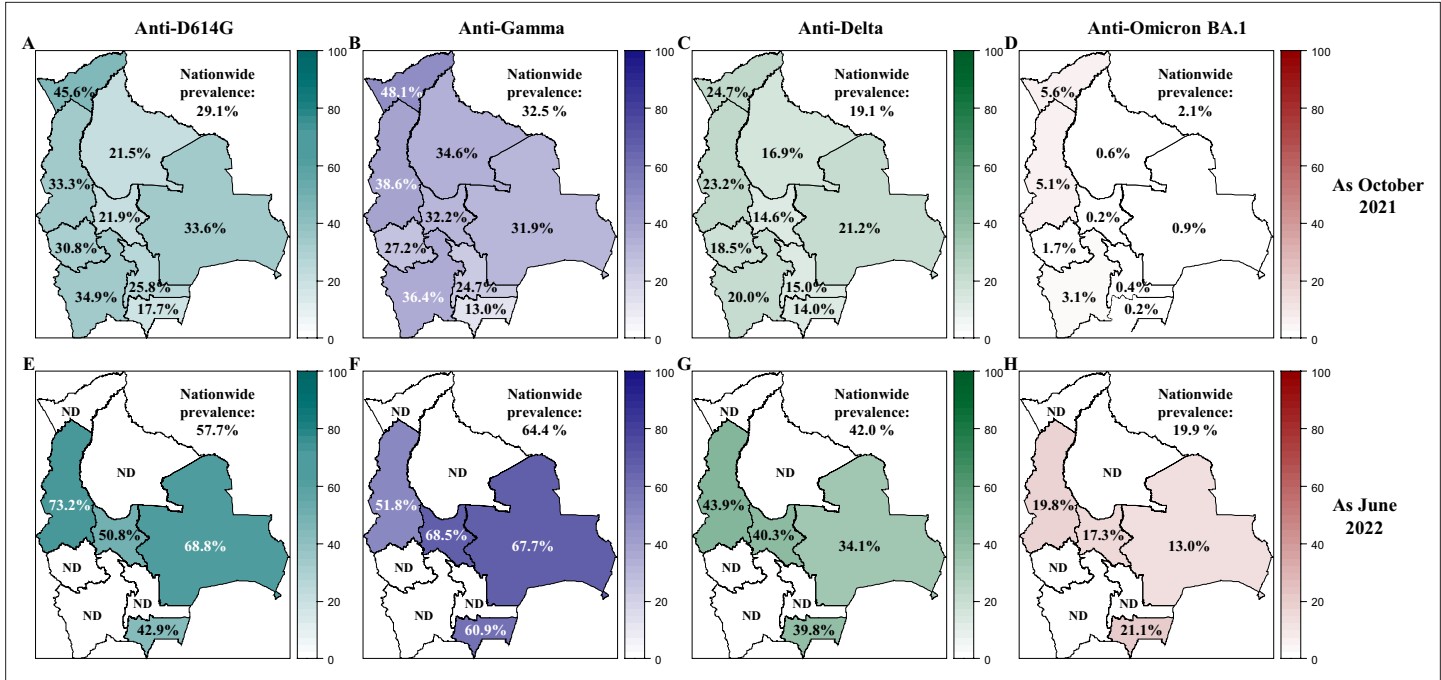

**Figure 2.** Prevalence of neutralizing anti-SARS-CoV-2 antibodies in Bolivia as of October 2021 and June 2022. The prevalence of neutralizing antibodies was reported for each department in Bolivia and nationwide for the D614G (**A, E**), Gamma (**B, F**), Delta (**C, G**) and Omicron BA.1 (**D, H**) variants as October 2021 (upper panel: **A, B, C, D**) and June 2022 (lower panel: **E, F, G, H**). The prevalence was obtained with the positivity cut-off ≥640.

The online version of this article includes the following figure supplement(s) for figure 2:

**Figure supplement 1.** Prevalence of neutralizing anti-SARS-CoV-2 antibodies in Bolivia as of October 2021 and June 2022.

**Figure supplement 2.** GMT titers of neutralizing anti-SARS-CoV-2 antibodies in Bolivia as of October 2021 and June 2022.

## Prevalence of nAbs against Gamma

In October 2021, the nationwide prevalence of Gamma nAbs at cut-offs ≥20 and ≥640 was 80.8% and 32.5% respectively, with a GMT of 208, showing a substantial circulation of Gamma, as expected. The three northern departments (Pando, La Paz, and Beni) had the highest GMTs, varying between 310 and 352. The lowest and highest values were observed in Tarija (63.7%) and Potosí (91.3%), respectively. Using the 'high-level cut-off' (≥640), similar to D614G, the highest prevalence was in Pando (48.1%), and the lowest in Tarija (13.0%).

By June 2022, the nationwide prevalence increased to 97.9% and 64.4% for the regular and high-level cut-offs, respectively, with a GMT of 778. The prevalence for the regular cut-off was above 96.5% across all tested departments. The highest and lowest prevalence values with the high-level cut-off were observed in Cochabamba (68.5%) and La Paz (51.8%), respectively.

## Prevalence of nAbs against Delta

In October 2021, despite expectations of low or no circulation of the Delta variant, we found 76.7% and 19.1% prevalence values at cut-offs ≥20 and ≥640 for Delta nAbs, respectively. This reflects the high titers observed for D614G, as the result of cross-neutralization within antigenic Group I. The Delta GMT was 81, compared to D614G (159) and Gamma (208).

By June 2022, the nationwide prevalence increased to 96.9% and 42.0% for the cut-offs ≥20 and ≥640, respectively, with a GMT of 230. The prevalence obtained with the cut-off ≥20 exceeded 86.7% across all tested departments, and the highest and lowest prevalence values with the cut-off ≥640 were found in La Paz (43.9%) and Santa Cruz (34.1%), respectively.

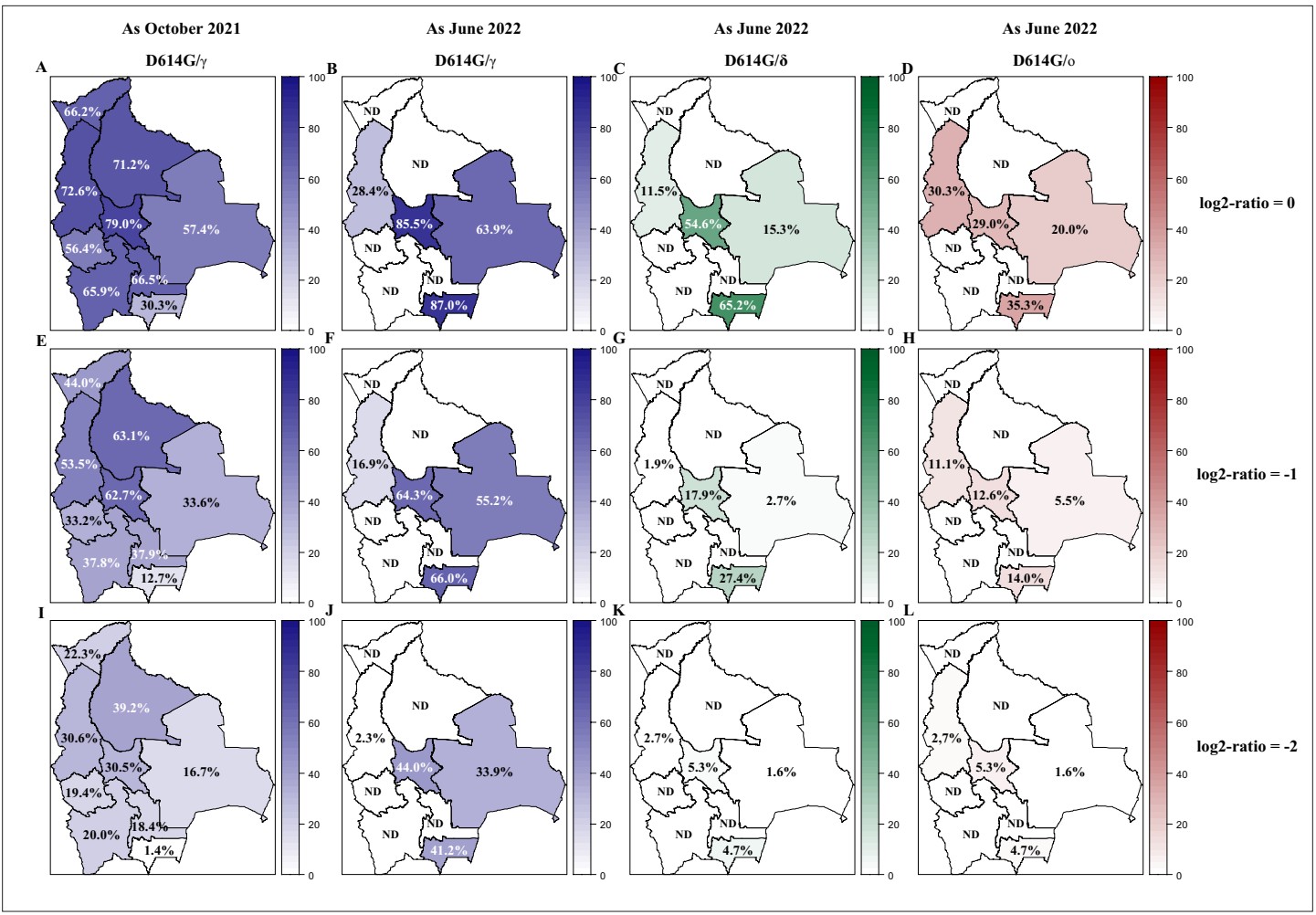

**Figure 3.** Variant circulation in Bolivia as of October 2021 and June 2022. The prevalence of neutralizing antibodies was reported for each department in Bolivia and nationwide for the D614G (**A, E**), Gamma (**B, F**), Delta (**C, G**), and Omicron BA.1 (**D, H**) variants as of October 2021 (upper panel: **A, B, C, D**) and June 2022 (lower panel: **E, F, G, H**). The prevalence was obtained with the positivity cut-off ≥640. Percentage of the population exhibiting titers for Gamma, Delta, and Omicron BA.1 being at least equivalent to (log2-ratios≤0, upper panel: **A, B, C, D**), twice (log2-ratios ≤ −1, middle panel: **E, F, G, H**), or four times (log2-ratios ≤ −2, lower panel: **I, J, K, L**) that of D614G are presented for each department of Bolivia. Results for D614G/γ are depicted in blue as October 2021 (**A, E, I**) and June 2022 (**B, F, J**). Results obtained as June 2022 for D614G/δ and for D614G/ο are depicted in green (**C, G, K**) and in red (**D, H, L**), respectively. D614G: ancestral D614G variant; γ: Gamma variant; δ: Delta variant; ο: Omicron BA.1 variant.

The online version of this article includes the following figure supplement(s) for figure 3:

**Figure supplement 1.** Flowchart for the identification of circulating variants.

**Figure supplement 2.** Variant circulation in Bolivia as of October 2021.

## Prevalence of nAbs against Omicron BA.1

In October 2021, prevalence for Omicron BA.1 was low (37.9% and 2.1% for the cut-off ≥20 and ≥640, respectively), with a GMT of 23. In the absence of Omicron BA.1 circulation at this time, this prevalence reflects cross-neutralization with D614G and Gamma.

By June 2022, nationwide prevalence increased to 90.7% and 19.9% for the cut-offs ≥20 and ≥640, respectively, with a GMT of 127. The highest prevalence at the cut-off ≥640 was recorded in Tarija (21.1%), and the lowest in Santa Cruz (13.0%).

## Evaluation of the circulating variants

This was performed using the 'VNT Titers ratio method' (see Methods for details). Increasingly stringent conditions (percentage of individuals in the population with titers of variants equivalent to, twice or four times that of D614G) were tested (**Figure 3**, **Figure 3—figure supplement 2**).

## Circulation of Gamma

In October 2021, in the northern part of Bolivia, encompassing the Pando, La Paz, Beni, and Cochabamba departments, the population exhibited the highest percentage of individuals with Gamma titers superior to D614G. Conversely, Tarija (South) showed the lowest percentage. This trend remained consistent across all conditions, including the most stringent ones.

By June 2022, the Gamma variant was more actively circulating in Tarija, Cochabamba, and Santa Cruz. Compared to the first sampling period, this variant was less active in La Paz.

We conclude that the gamma variant circulated mainly in the north of Bolivia before October 2021, then spread to the southern regions of the country.

## Circulation of Delta and Omicron BA.1

In October 2021, evidence for the circulation of the Delta and Omicron BA.1 variants was minimal, as expected (*Figure 1—figure supplement 1*; *PAHO, 2023*; *GISAID, 2023*; *Hodcroft, 2023*).

By June 2022, in contrast, the departments tested displayed an increase in the percentage of the population with Delta and Omicron BA.1 titers superior to D614G. The highest values for both Delta and Omicron BA.1 were found in Tarija and Cochabamba, and the lowest in La Paz and Santa Cruz.

We conclude that there is serological evidence for the circulation of both Delta and Omicron BA.1 variants after October 2021.

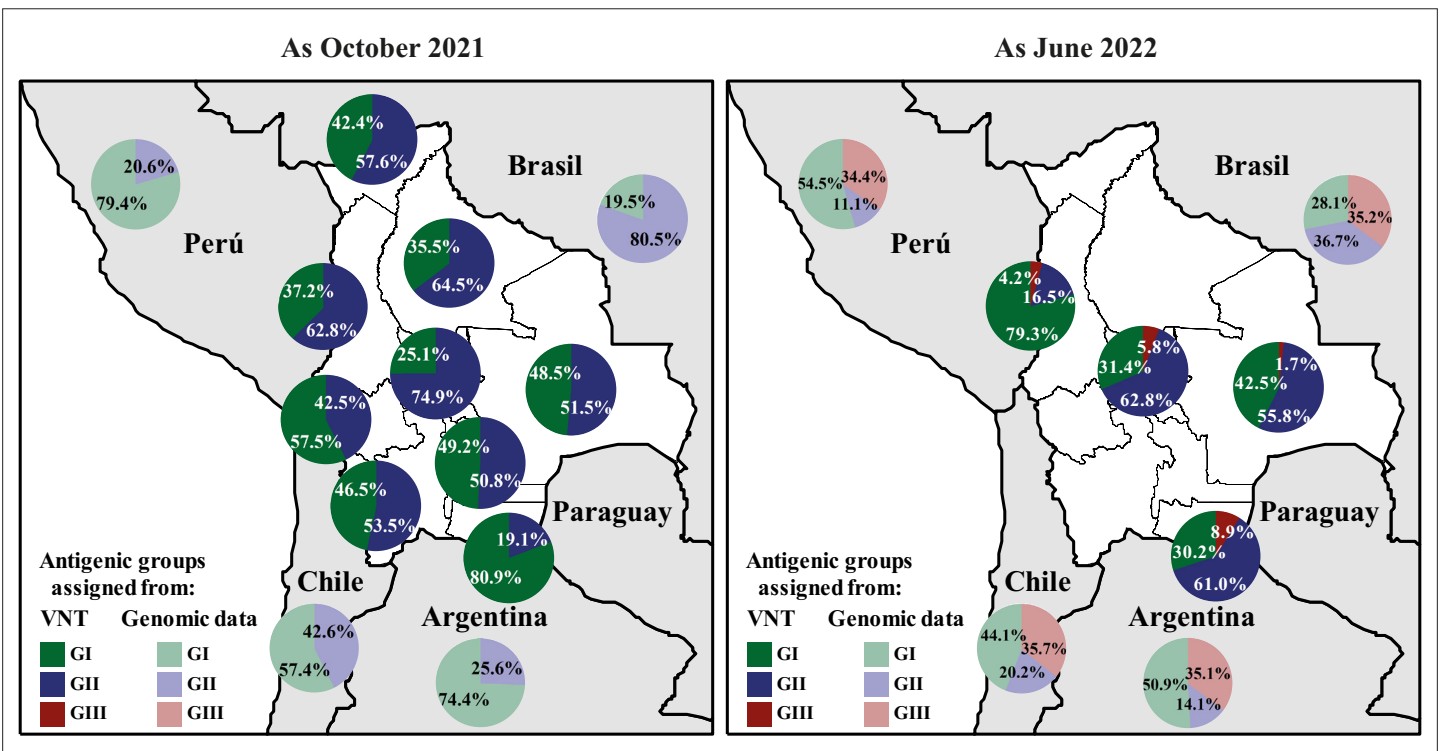

**Figure 4.** Quantification of SARS-CoV-2 circulating antigenic groups in Bolivia by Average titer method as October 2021 and as June 2022. Percentage of the population in each department presenting a neutralizing antibody response against a defined antigenic group, including the GI (green), GII (blue), GIII (red), using the Average titer method. For antigenic group assignment by VNT, D614G and Delta variants were clustered into the antigenic group I (GI), antigenic group II (GII) corresponded to the Gamma variant, while antigenic group III (GIII) was defined by the Omicron BA.1 variant (see Methods for details). For evaluating genomic data, the prevalences of GI (from ancestral, Alpha, Delta, Lambda, and Epsilon variants; light-green), the prevalences of GII (from Beta, Gamma, Mu, Iota, and Zeta variants; light-blue), and the prevalences of GIII (from all Omicron-derived variants; light-red) were obtained for neighboring countries of Bolivia using publicly accessible genomic data when available.

The online version of this article includes the following figure supplement(s) for figure 4:

**Figure supplement 1.** Quantification of SARS-CoV-2 circulating antigenic groups in Bolivia by Variant assignment method as October 2021 and as June 2022.

**Figure supplement 2.** Quantification of SARS-CoV-2 circulating antigenic groups in Bolivia by Rules method as October 2021 and as June 2022.

### Quantification of the circulating antigenic groups

Using two approaches (the 'Average titer method' and the 'Variant assignment method'), the circulating antigenic groups were quantified (*Figure 4*, *Figure 4—figure supplement 1*). The nAbs responses resulting mainly from vaccination, ancestral, and Delta variants were clustered in group I (GI), and Gamma and Omicron BA.1 were the representatives of groups II (GII) and III (GIII), respectively. These two approaches yielded highly congruent results that were also in agreement with the above-mentioned results (*Figure 3*). Data from the 'Average titer method' was exposed in the following section (*Figure 4*).

### Antigenic group I

In October 2021, the percentages of GI-associated nAbs (GI-nAbs) spanned from 25.1% (Cochabamba) to 80.9% (Tarija). The southernmost departments (Oruro, Potosí, and Tarija) exhibited the highest values, aligning with genomic prevalence from surrounding countries for which GI variants circulated more in Argentina (74.4%), Chile (57.4%), and Perú (79.4%).

By June 2022, the percentages of GI-nAbs extended from 30.2% (Tarija) to 79.3% (La Paz). Compared to October 2021, the departments of La Paz and Cochabamba showed an increase in the levels, while Santa Cruz and Tarija exhibited a decrease.

### Antigenic group II

In October 2021, the percentages of GII-associated nAbs (GII-nAbs) spanned from 19.1% (Tarija) to 74.9% (Cochabamba). The northern departments (Pando, La Paz, Beni, and Cochabamba) presented the highest levels, which reflects the circulation in neighboring Brazil (80.5%).

By June 2022, the percentages of GII-associated nAbs (GII-nAbs) spanned from 16.5% (La Paz) to 62.8% (Cochabamba). The southern departments of Santa Cruz and Tarija showed an increase, while the northern departments of La Paz and Cochabamba showed a decrease in this response.

### Antigenic group III

In October 2021, no evidence of circulation of GIII variants was observed from GIII-associated nAbs (GIII-nAbs) in Bolivia, or genomic surveillance derived data from surrounding countries.

By June 2022, the percentages of GIII-nAbs varied from 1.7% (Santa Cruz) to 8.9% (Tarija). GIII-nAbs increased across all tested departments, i.e. the method was capable of detecting the emergence of the new antigenic group.

We conclude that these serological data nicely align with the circulation of each antigenic group in the neighboring countries as computed from accessible genomic surveillance data, particularly at the first sampling period (*Figure 4*, *Figure 1—figure supplement 1*).

## Quantification of the individual variant circulation

The prevalence of individual variants, assigned through the Assignment Rules, was examined across all departments, showing similar trends with the above methods but offering a more accurate assessment of ancestral and Delta variant circulation and possible insights into the weight of vaccination in the generation of ancestral D614G-induced nAbs antibodies (*Figure 3—figure supplement 2*, *Figure 4—figure supplement 1*).

### Quantification of ancestral D614G-related variants circulation

In October 2021, Tarija exhibited the highest D614G prevalence, while the northern departments had some of the lowest circulation rates, a pattern consistent with vaccination coverage data (*Figure 1—figure supplement 2*) and the results observed above.

By June 2022, the southernmost departments of Santa Cruz and Tarija saw a reduction in D614G prevalence. This trend comes with a higher level of Gamma circulation observed in these two departments (see above), further supporting the hypothesis of ongoing Gamma circulation between 2021 and 2022. In the department of Cochabamba, no significant change in D614G prevalence was observed (28.3% in October 2020 to 26.6% in June 2022). In contrast, La Paz experienced a notable surge (+40%) in D614G prevalence, consistent with the antigenic group analysis.

Quantification of Delta circulation

In October 2021, a low circulation of Delta was observed in all departments, ranging from 1.1% (Pando) to 9.1% (Tarija). As a result, the method was able to detect the appearance of the Delta variant, despite its belonging to the same antigenic group as other variants circulating previously.

By June 2022, the prevalence of Delta remained stable or decreased across the tested departments, with prevalences ranging between 1.5% (La Paz) and 9.1% (Tarija).

## Discussion

To our knowledge, we are reporting the first serological study of SARS-CoV-2 to be carried out in Bolivia in 2021 and 2022, which included blood donors from all the country's departments. This study enabled us to estimate the level of immunity of the population to SARS-CoV-2 nationwide and in the different regions of the country, thus providing valuable information on the circulation of these variants in Bolivia.

### Limitations

Firstly, the study was limited to blood donors in the major cities of Bolivia and may therefore not be fully representative of the populations of the country as a whole. Furthermore, in Bolivia, blood donation is unrewarded, and blood donors appear to be quite representative of the general population. Indeed, routine screening for several infection markers (such as HIV or HBV) is conducted in all donors, and the prevalences of these markers do not differ from those observed in the general population. For example, UNAIDS data highlights a 0.4% HIV prevalence within the Bolivian general population, with significantly higher rates exceeding 25% observed in high-risk groups such as men who have sex with men (*UNAIDS, 2022*). Moreover, *Sheena et al., 2022* estimated a 0.6% prevalence of HBsAg in Bolivia in 2019. Bolivian national statistics of National Blood Program of the Ministry of Health and Sports indicate that between 2019 and 2023, the proportion of HIV- and HBV-reactive units among screened blood donors ranged from 0.26% to 0.41% and 0.16% to 0.25%, respectively (Dr. Lissete Bautista's personal communication).

Secondly, our results are based on serological data and may not be strictly identical to the genomic data from a quantitative point of view, although they are likely to reflect similar trends and distributions (see below). The results could also be influenced by various factors, including significant individual variation in antibody responses, as well as the decline in antibody titers during the first months following infection or vaccination (*Iyer et al., 2020*; *Hartley et al., 2020*; *Turner et al., 2021*; *Peghin et al., 2021*) and could therefore be slightly underestimated. As the complexity of SARS-CoV-2 antigen exposure histories increased among tested individuals, we observed a tendency for serological data to start diverging from genomic data. This suggests, as expected, that the effectiveness of this method would be greater if implemented early in an epidemic when the occurrence of multiple infections with different variants or the administration of varying doses of vaccine in the analyzed population before or after infection (resulting in hybrid immunity) is still limited. However, to mitigate the potential challenges arising from complex antigen exposure, we employed straightforward criteria to identify the variant among the four tested in VNT that exhibited the highest value (cf methods), thereby likely indicating the main or most recent infection and minimizing the influence of cross-neutralization on the final outcomes. In addition, several approaches were used to analyze the results, including quantification of circulating antigenic groups and individual variants, yielding results that were comparable and closely aligned with the genomic data.

### Epidemiological assessment

As of October 2021, our data showed an anti-S1 SARS-CoV-2 seroprevalence of 86.3%, indicating that a large part of the population had immunity against SARS-CoV-2 following infection, vaccination, or both (*Figure 1*). Comparing the vaccination coverage with the anti-S1 antibody prevalence, we can infer that significant viral transmission occurred in the population before October 2021 (*Figure 1*, *Figure 1—figure supplement 2*). These epidemiological data are in agreement with other population-based studies performed on the first half of 2021 in Perú and in the northwest of Brazil that reported high SARS-CoV-2 prevalence (63.6% and 65.0%, respectively) (*Álvarez-Antonio et al., 2021*; *Buss et al., 2021*). In Bolivia, the prevalence observed a few months later was even

higher, suggesting sustained virus circulation. By June 2022, although our sample was limited to four departments, the data showed a clear trend towards saturation in the prevalence of anti-S1, suggesting that almost the entire population had been vaccinated and/or infected by that time. With vaccination coverage having increased by around 13% (both for those who had received two doses and those who had received the booster), the significant increase (+31.2%) in the seroprevalence of anti-NCP antibodies indicates that a large proportion of the population has been exposed to the virus between October 2021 and June 2022 (*Figure 1*). Even if a proportion of newly vaccinated individuals had received an inactivated vaccine likely to generate an anti-NCP response, this cannot explain the observed increase in the prevalence of anti-NCP antibodies (*Muena et al., 2022*; *Gao et al., 2020*; *Dashdorj et al., 2021*). This corresponds to the emergence of the Omicron BA.1 variant, which has been shown to cause a high rate of infection worldwide (*Silva et al., 2023*; *Karim and Karim, 2021*).

Consistently, as of October 2021, the prevalence of nAbs against the ancestral D614G (83.1%) and Gamma (80.8%) variants was very high throughout the country (*Figure 2*, *Figure 2—figure supplement 1*). The three northern departments (Pando, La Paz, and Beni) were the most exposed to the Gamma variant, as reflected by the highest Gamma GMTs (*Figure 2—figure supplement 2*). By June 2022, the prevalence of neutralizing antibodies against all variants had increased, reaching values higher than 90%, demonstrating that a large proportion of the Bolivian population was no longer naïve to the virus.

When looking at nAbs with high neutralization titers, new variant circulation patterns emerged (*Figure 2*, *Figure 2—figure supplement 1*). Some departments in the northern part of Bolivia, especially Pando, may have suffered a greater impact during the first two waves, since nAbs titers were higher for both the D614G and Gamma variants as of October 2021. To gain further insight into the variant circulation, we proposed additional analyses ('VNT Titers ratio method', 'Average titer method', 'Variant assignment method'; *Figures 3 and 4*, *Figure 3—figure supplement 2* and *Figure 4—figure supplements 1 and 2*).

As of October 2021, a higher circulation of GII/Gamma was confirmed in northern Bolivia. Moreover, it is likely that the first two waves had a significant impact on Pando, since (despite having the lowest vaccine coverage in the country) high percentages of nAbs responses to the ancestral D614G and Gamma variants were observed. On the other hand, the southernmost department (Tarija) was probably the least affected, as it had the highest percentage of neutralizing response to ancestral viruses and the lowest to Gamma, which is consistent with its high vaccination coverage (the highest in the country). As for Delta, limited circulation was observed across the country, corresponding to the beginning of its circulation in Latin America (*Figure 1—figure supplement 1*). As expected, no evidence of Omicron BA.1 was observed.

By June 2022, analyses showed that GII/Gamma continued to circulate in southernmost departments (Cochabamba, Santa Cruz, and Tarija). Besides, the high level of nAbs against GII/Gamma, observed in the first sampling period, in the population of La Paz seems to have waned, and on the contrary, GI/D614G nAbs have risen (+40%). This observation cannot be explained solely by an increase in vaccination coverage (*Figure 1—figure supplement 2*), since vaccination increased by only 13%. GI/D614G nAbs augmentation could be related to the circulation of another GI variant, such as Lambda (which has circulated in the neighboring country of Perú), or alternatively, the infection by the Delta and/or Omicron BA.1 variant may have boosted the level of pre-existing anti-D614G nAbs in the population (*Quandt et al., 2022*; *Kaku et al., 2022*; *Stamatatos et al., 2021*; *Gallian et al., 2023*). This boost of pre-existing Abs phenomenon, known as antigenic sin, has been extensively documented in the context of influenza infections and vaccinations (*Viboud and Epstein, 2016*; *Monto et al., 2017*). As for Delta, prevalences did not change significantly between October 2021 and June 2022, reflecting the limited Delta wave in Latin America compared to Europe (*Giovanetti et al., 2022*). Despite a significant global wave of the Omicron BA.1 variant in South America, less than 9% of the Bolivian population tested displayed a neutralizing antibody response against Omicron BA.1. This lower-than-expected nAbs response could be attributed to the possibility that this variant increased the levels of pre-existing nAbs to previous variants (*Quandt et al., 2022*; *Kaku et al., 2022*; *Stamatatos et al., 2021*; *Gallian et al., 2023*).

## Significance of serological results

To assess the significance and robustness of our serological results, we compared the results obtained with the different methods used and with genomic data from neighboring countries. We found that the patterns of circulation of antigenic groups/variants were consistent across the different serological methods used (*Figure 4*, *Figure 4—figure supplements 1 and 2*) and that they nicely aligned with the percentage of each antigenic group/variants in the neighboring countries as computed from accessible genomic surveillance data (*Figure 1—figure supplement 1*).

Serological analyses allowed detecting the emergence of a new antigenic group (e.g., Omicron BA.1), but also of an individual variant related to a previously circulating antigenic group (e.g. Delta). Overall, the accuracy of the mapping seems to have diminished with time, the number of variants that have been circulated, vaccination, and the saturation of prevalence estimates. As an example, the prevalence values found for Omicron BA.1 in June 2022 are likely to be underestimated (see *Figure 4*, *Figure 4—figure supplements 1 and 2*). Although the detection of a new antigenic group remained possible in this specific study during the late period of variants circulation, our results argue in favor of the greater usefulness of seroepidemiological studies for documenting the early stages of a new epidemic. This is of great interest because it gives time to setup genomic surveillance in countries with limited sequencing capacity at the start of an epidemic emergence.

In conclusion, our study represents the first large-scale survey conducted in all the departments of Bolivia, with a follow-up period of almost a year. This has enabled us to document ongoing viral circulation and better characterize the epidemiology of SARS-CoV-2 in Latin America, as well as providing essential information on the successive waves that have occurred in Bolivia. In particular, despite a high prevalence in 2022, which could suggest the development of herd immunity, subsequent waves demonstrate the virus's ability to adapt and continue circulating in a population that is no longer naïve. Our results confirm and reiterate that seroprevalence studies are a powerful tool for informing public health decision-making. Our findings strongly support the idea that seroepidemiological studies are valuable for capturing the initial phases of a new epidemic, giving time for the establishment of genomic surveillance in countries with limited sequencing capacity at the onset of an emerging epidemic.

## Ethical approval

Samples were collected randomly from volunteer blood donors who agreed to participate in the study. Written consent was obtained from all blood donors. The study was approved by the ethics committee of the Dr. Mario Ortíz Suárez hospital, Santa Cruz de la Sierra, Bolivia (N° FWA0002686) and by the National Blood Program from the Ministry of Health and Sports of Bolivia. Sampling was carefully conducted in close collaboration with the National Blood Program and each Departmental Reference Blood Bank.

## Acknowledgements

The authors thank all professionals from the Departmental Reference Blood Banks. A special thanks to Dra. Daniela Azurduy, Dra. Vanessa Telleria, Dra. Cecilia Peralta, Dra. Rosario Quintanilla, Dra. Rosa Escalier, Dra. Patricia Achacollo Olmo, Dra. Raquel Verónica Figueredo Flores, Dr.Ronald Flores Zabala, and Dra. Maria Luisa Patón for their support and management to carry out the sampling for the project. We would also like to thank the collaboration of the Ministerio de Salud y Deportes, the Viceministerio de Gestión del Sistema de Salud and the Dirección de Redes de Servicios de Salud. This study was supported by the French National Research Institute for Sustainable Development (IRD), the project EMERGEN-PRI #22275 of the ANRS I MIE (INSERM), and the European Union's Horizon 2020 research and innovation program (European Virus Archive Global, grant agreement No. 871029). The funders of the study had no role in study design, data collection, data analysis, data interpretation, or writing of the report.

## Additional information

### Funding

| Funder | Grant reference number | Author |
|---|---|---|
| French National Research Institute for Sustainable Development | ARTS | Lucia Inchauste |
| ANRS MIE | EMERGEN-PRI #22275 | Xavier de Lamballerie |
| Horizon 2020 Framework Programme | EVAg #871029 | Xavier de Lamballerie |

The funders had no role in study design, data collection and interpretation, or the decision to submit the work for publication.

### Author contributions

Lucia Inchauste, Conceptualization, Resources, Data curation, Formal analysis, Validation, Investigation, Visualization, Methodology, Writing – original draft, Writing – review and editing; Elif Nurtop, Methodology; Lissete Bautista Machicado, Yanine Leigue Roth, Shirley Lenz Gonzales, Maria Luisa Herrera, Katty Mina Villafan, Pedro Mamani Mamani, Marcelo Ramos Espinoza, Juan Carlos Pavel Suarez, Juan Cansio Garcia Copa, Yitzhak Leigue Zabala, Etzel Arancibia Cardozo, Pierre Gallian, Resources; Xavier de Lamballerie, Funding acquisition, Writing – review and editing; Stéphane Priet, Conceptualization, Data curation, Formal analysis, Supervision, Validation, Investigation, Methodology, Writing – original draft, Project administration, Writing – review and editing

### Author ORCIDs

Stéphane Priet ⦿ https://orcid.org/0000-0001-7102-3654

### Ethics

Samples were collected randomly from volunteer blood donors that agreed to participate in the study. Written consent was obtained from all blood donors. The study was approved by the ethics committee of the Dr. Mario Ortz Suárez hospital, Santa Cruz de la Sierra, Bolivia (N° FWA0002686) and by the National Blood Program from the Ministry of Health and Sports of Bolivia. Sampling was carefully conducted in close collaboration with the National Blood Program and each Departmental Reference Blood Bank.

Reviewer #1 (Public review): https://doi.org/10.7554/eLife.94475.3.sa1
Reviewer #3 (Public review): https://doi.org/10.7554/eLife.94475.3.sa2
Author response https://doi.org/10.7554/eLife.94475.3.sa3

## Additional files

### Supplementary files

MDAR checklist

Supplementary file 1. SARS-CoV-2 hybrid immunity estimation by department in 2021 and 2022.

Supplementary file 2. Serological tests raw data.

### Data availability

All data generated or analysed during this study are included in the manuscript and supporting files.

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
