## [Editor Report · eLife Assessment]

This serostudy of blood donors in Bolivia (a country with very high COVID death rates in 2020-21) provides **useful** insights on the successive viral variants of SARS-CoV-2 over 2021 and 2022. Using **compelling** antibody and neutralization assays, the authors describe variant specific distributions in the different parts of Bolivia. The main methodological advance is to use serology to understand variant diversity, which in turn helps deepen understanding of "hybrid" immunity from widespread infection (and vaccination).

---

## [Referee Report · Reviewer #1 (Public review)]

Summary:

This study provides valuable and comprehensive information about the SARS-CoV-2 seroprevalence during 2021 and 2022 in different regions of Bolivia. Moreover, data on immune responses against the SARS-CoV-2 variants based on neutralization tests denotes the presence of several virus variants circulating in the Bolivian population. Evidence for seroprevalence data provided by the authors is solid, across the study period, while data regarding variant circulation is limited to the early stages of the pandemic.

Strengths:

The major strength of this study is that it provided nationwide seroprevalence estimates from infection and/or vaccination based on antibodies against both spike and the nucleocapsid protein in a large representative sample of sera collected at two time points from all departments of Bolivia, gaining insight into COVID-19 epidemiology. On the other hand, data from virus neutralization assays inferred the circulation during the study period of four SARS-CoV-2 variants in the population. Overall, the study results provide an overview of the level of viral transmission and vaccination and insights into the spread across the country of SARS-CoV-2 variants.

Weaknesses:

The assessment of a Lambda variant that circulated in several neighboring countries (Peru, Chile, and Argentina), which had a significant impact on the COVID-19 pandemic in the region, may have strengthened the study to contrast Gamma spread. In addition, even though neutralizing antibodies can certainly reveal previous infections of SARSCOV2 variants in the population, it is of limited value to infer from this information some potential timing estimates of specific variant circulation, considering the heterogeneous effects that past infections, vaccinations, or a combination of both could have on the level of variant-specific neutralizing antibodies and/or their cross-neutralization capacity.

An appraisal of whether the authors achieved their aims, and whether the results support their conclusions.

The conclusions of this paper are well supported by data, particularly regarding seroprevalence that reliably reflects the epidemiology of COVID-19 in Bolivia, and seroprevalence trends in other low- and middle-income countries.

A discussion of the likely impact of the work on the field, and the utility of the methods and data to the community.

Since this is the first study that has been conducted to assess indicators of immunity against SARS-CoV-2 in the population of Bolivia at a nationwide scale, seroprevalence data provided by geographic regions at two time points can be useful as a reference for potential retrospective global meta-analysis and to further explore and compare the risk factors for infection, variant distribution, and the impact on infection and vaccination, gaining deeper insights into understanding the evolution of the COVID-19 pandemic in Bolivia and in the region.

---

## [Referee Report · Reviewer #3 (Public review)]

Summary:

This study attempts to reconstruct the history of the COVID-19 epidemic, with its successive waves of viral variants from SARS-CoV-2 seroprevalence during 2021 and 2022 among blood donors in different regions of Bolivia. By using serological tests "specific" for the various variants the authors try to achieve a "colour" vision that is not provided by standard "black-and-white" serology.

Strengths and Weaknesses:

I am not an expert on the performance of SARS-CoV-2 serological tests, so may overlook certain weaknesses. Instead I tried to assess whether the authors, in this manuscript, have managed to substantiate their claims that "seroprevalence studies are a valuable adjunct to active surveillance because they allow analysis of the level of immunity of a population to a specific pathogen without the need for prospective testing" , and that "genomic surveillance and serology offer distinct yet complementary insights thus far." I think they succeeded, as they paint a credible and interesting history of the epidemic in Bolivia using (to me) novel methodology that certainly will stimulate extensive discussion, controversies, and follow-up studies (for which the authors might make some suggestions).

---

## [Author Response]

The following is the authors’ response to the original reviews.

**Reviewer #1 (Public Review):**
Summary:This study provides valuable and comprehensive information about the SARS-CoV-2 seroprevalence during 2021 and 2022 in different regions of Bolivia. Moreover, data on immune responses against the SARS-CoV-2 variants based on neutralization tests denotes the presence of several virus variants circulating in the Bolivian population. Evidence for seroprevalence data provided by the authors is solid, across the study period, while data regarding variant circulation is limited to the early stages of the pandemic.Strengths:The major strength of this study is that it provided nationwide seroprevalence estimates from infection and/or vaccination based on antibodies against both spike and the nucleocapsid protein in a large representative sample of sera collected at two time-points from all departments of Bolivia, gaining insight into COVID-19 epidemiology. On the other hand, data from virus neutralization assays inferred the circulation during the study period of four SARS-CoV-2 variants in the population. Overall, the study results provide an overview of the level of viral transmission and vaccination and insights into the spread across the country of SARS-CoV-2 variants.Weaknesses:The assessment of a Lambda variant that circulated in several neighboring countries (Peru, Chile, and Argentina), which had a significant impact on the COVID-19 pandemic in the region, may have strengthened the study to contrast Gamma spread. In addition, even though neutralizing antibodies can certainly reveal previous infections of SARSCOV2 variants in the population, it is of limited value to infer from this information some potential timing estimates of specific variant circulation, considering the heterogeneous effects that past infections, vaccinations, or a combination of both could have on the level of variant-specific neutralizing antibodies and/or their cross-neutralization capacity.An appraisal of whether the authors achieved their aims, and whether the results support their conclusions:The conclusions of this paper are well supported by data, particularly regarding seroprevalence that reliably reflects the epidemiology of COVID-19 in Bolivia, and seroprevalence trends in other low- and middle-income countries.A discussion of the likely impact of the work on the field, and the utility of the methods and data to the community:Since this is the first study that has been conducted to assess indicators of immunity against SARSCoV-2 in the population of Bolivia at a nationwide scale, seroprevalence data provided by geographic regions at two time-points can be useful as a reference for potential retrospective global metaanalysis and further explore and compare the risk factors for infection, variant distribution, and the impact on infection and vaccination, gaining deeper insights into understanding the evolution of the COVID-19 pandemic in Bolivia and in the region.
**Reviewer #2 (Public Review):**
Significance of the findings:In this study, blood donors were assessed using serology and viral neutralization assays to determine the prevalence of SARS-CoV-2 antibodies. S1 and NCP antibodies were used to distinguish between vaccination and natural infection and virus-specific neut titers were used to determine which variants the antibodies respond to. The study reports almost universal antibody prevalence and increases in antibodies against specific variants at different points corresponding to circulating variants identified phylogenetically in neighbouring countries. The authors propose this approach for settings like Bolivia where genetic sequencing is not readily available. Unfortunately, there are significant limitations to this approach that limit its utility - serological data are available after the fact in a fast-moving pandemic and so are a poor alternative to phylogenetic data. Rather, serological information can supplement phylogenetic data and is most useful in estimating population-level immunity.(1) Considerations in interpreting the results:

We appreciate the reviewer's valuable feedback, which will certainly enhance the quality of our manuscript. As a result, we have revised the text to address their suggestions as thoroughly as possible.

a. Serology provides different information to phylogenetic sequencing of the viruses and so both are important. Viral sequencing provides real-time information on circulating variants and indicates the proportion of each variant in circulation at any point as there are almost always multiple variants spreading but it is the fastest spreading variant that comes to dominate. Importantly serology measures asymptomatic infections as well, providing population estimates of infection that are not available through viral gene sequencing.

We underscored this point in the introduction by incorporating the following sentences:

“Seroprevalence studies are a valuable adjunct to active surveillance because they allow analysis of the level of immunity of a population to a specific pathogen without the need for prospective testing, and also provide information on the frequency of cases that do not attract medical attention (asymptomatic infections)(4).” and “To date, the circulation of SARS-CoV-2 variants has mainly been studied through molecular surveillance, giving the proportion of circulating variants in real time. Therefore, genomic surveillance and serology offer distinct yet complementary insights thus far.”

b. A major concern in the interpretation of serology is that antibody titers vary markedly over time with rapid declines in the first year post-infection or post-vaccination. However, these declines vary depending on whether hybrid immunity is present. Disentangling this retrospectively is a challenge. A low antibody titer could reflect an infection that occurred a few months ago but may be below the threshold for positivity at the time of testing. There is also substantial individual variability in antibody responses.

This limitation merits emphasis and has consequently been elaborated upon in the discussion section:

“Secondly, our results are based on serological data and may not be strictly identical to the genomic data from a quantitative point of view, although they are likely to reflect similar trends and distributions (see below). The results could also be influenced by various factors, including significant individual variation in antibody responses, as well as the decline in antibody titers during the first months following infection or vaccination(31-34) and could therefore sligly underestimated. As the complexity of SARS-CoV-2 antigen exposure histories increased among tested individuals, we observed a tendency for serological data to start diverging from genomic data. This suggests, as expected, that the effectiveness of this method would be greater if implemented early in an epidemic when the occurrence of multiple infections with different variants or the administration of varying doses of vaccine in the analyzed population before or after infection (resulting in hybrid immunity) is still limited. However, to mitigate the potential challenges arising from complex antigen exposure, we employed straightforward criteria to identify the variant among the four tested in VNT that exhibited the highest value (cf methods), thereby likely indicating the main or most recent infection and minimizing the influence of crossneutralization on the final outcomes. In addition, several approaches were used to analyze the results, including quantification of circulating antigenic groups and individual variants, yielding results that were comparable and closely aligned with the genomic data.”

c. Serology becomes increasingly difficult to untangle when an individual has had doses of vaccine and multiple natural infections with different variants. Due to the importance of hybrid immunity in population risk to new variants, it would be useful for estimates of hybrid immunity to be generated based on anti-S1 and anti-NCP antibodies. From a population immunity perspective, this could be important in guiding future protection and boosting strategies.

We estimated the hybrid immunity for each department in 2021 and 2022 based on the prevalence of anti-S1 and anti-NCP antibodies and added a new Supplementary Table 1. We also added a description of this table in the result section: “The estimated hybrid immunity, based on the prevalence of anti-S1 and anti-NCP antibodies, ranged from 51.4% in Pando to 73.6% in Potosí in 2021. By 2022, this increased to between 83.3% in Santa Cruz and 90.6% in Tarija (Supplementary Table 1).”

d. Since there is cross-neutralization by the antibodies stimulated by each variant, it is important to establish the sensitivity and specificity of each of the neutralization assays in a panel comprising multiple variants. An assessment of the accuracy of the neut assay for each variant is needed to be confident that it is able to distinguish between variants.

Assessing the performance of a the VNT for each SARS-CoV-2 variants is a highly complex task. This evaluation requires samples with comprehensive data on vaccination and infection specific to each variant to determine the specificity of each VNT for each variant. However, the access to such samples for every newly emerging variant remains challenging. In order to circumvent this issue, we evaluated the circulation level of γ, δ, and ο variants under increasingly stringent conditions, by calculating the proportion of the population with log2-ratio values of ≤0 (variant titer equal to or greater than D614G), ≤-1 (variant titer at least twice that of D614G), and ≤-2 (variant titer at least four times that of D614G).

e. Blood donors are notoriously poor representations of the general population in many countries, driven partly by whether donation is financially rewarded. For example, in the USA, drug addicts are disproportionately over-represented in blood donor populations as they use it as a source of money. The authors provide no information on whether the blood donor population in Bolivia is representative of the entire population. Comparison of the prevalence of specific disease markers in the general population and in blood donors could provide a signal of their comparability.

This is a significant aspect addressed in point 3.

(2) Please provide the sensitivity and specificity of each of the assays so that the reader can assess the degree of accuracy in the assay that claims that the prevalent antibodies are due to, for example, omicron.

The sensitivity and specificity of the in vitro assays are now referenced in a previous study: “The sensitivity and specificity of the in vitro assays were described previously(23).”

Neutralization assays are considered the gold standard for measuring neutralizing antibodies against SARS-CoV-2 and its variants, and they are widely used in seroprevalence studies. However, until now, no one has successfully evaluated the specificity and sensitivity of this assay for SARS-CoV-2 variants, as it requires sera from individuals exposed to a single variant, which are increasingly difficult to collect for each newly emerging variants. Nevertheless, using sera from laboratory-infected animals (primarily hamsters) with a single variant exposure has enabled the antigenic characterization of SARS-CoV-2 variants through viral neutralization. This approach has shown that it is possible to distinguish between sera from individuals infected with different variants, even among the Omicron subvariants (Anna Z. Mykytyn et al. Antigenic cartography of SARS-CoV-2 reveals that Omicron BA.1 and BA.2 are antigenically distinct.Sci. Immunol.7,eabq4450(2022); Samuel H. Wilks et al. Mapping SARS-CoV-2 antigenic relationships and serological responses.Science382,eadj0070(2023)).

(3) Please provide an assessment of the representativity of the blood donor population eg. Is the prevalence of hepatitis B serological markers in the blood donor population comparable with the prevalence of hepatitis B serological markers in the general population from community-based studies?

A new sentence was included in the discussion to offer support for considering the blood donor population as a representative sample of the general population: “In addition, in Bolivia, blood donation is unrewarded, and blood donors appear to be quite representative of the general population. Indeed, routine screening for several infection markers (such as HIV or HBV) is conducted in all donors, and the prevalences of these markers do not differ from those observed in the general population. For example, UNAIDS data highlights a 0.4% HIV prevalence within the Bolivian general population, with significantly higher rates exceeding 25% observed in high-risk groups such as men who have sex with men(29). Moreover, Sheena et al. estimated a 0.6% prevalence of HBsAg in Bolivia in 2019(30). Bolivian national statistics of National Blood Program of the Ministry of Health and Sports, indicate that between 2019 and 2023, the proportion of HIV- and HBV-reactive units among screened blood donors ranged from 0.26% to 0.41% and 0.16% to 0.25%, respectively (Dr. Lissete Bautista’s personal communication).”